# Neutralization of the Surface Charge of an Insulated Target under the Interaction of High-Energy Metal Ion Beams

**Konstantin P. Savkin [1,*], Efim M. Oks [1,2], Alexey G. Nikolaev [1] and Georgy Yu. Yushkov [1]**

[1] Institute of High Current Electronics SB RAS, 2/3, Akademicheskii Ave., 634055 Tomsk, Russia;
oks@opee.hcei.tsc.ru (E.M.O.); nik@opee.hcei.tsc.ru (A.G.N.); gyushkov@mail.ru (G.Y.Y.)

[2] Department of Physics, Faculty of Electronic Engineering, Tomsk State University Control Systems and Radioelectronics, 40, Lenin Ave., 634050 Tomsk, Russia

[*] Correspondence: savkin@opee.hcei.tsc.ru; Tel.: +7-913-856-83-57

**Abstract:** The interaction of ion beams with dielectric materials is an urgent problem, both from the point of view of practical application in ion implantation processes and for understanding the fundamental processes of charge compensation and the effective interaction of beam ions with a target surface. This paper presents the results of studies of the processes of compensation of the surface charge of an insulated collector upon interaction with a beam of metal ions with energies up to 50–150 keV. At low pressure (about $10^{-6}$ torr), removing the collector from the region of extraction and beam formation makes it possible to reduce the floating potential to a value of 5–10% of the total accelerating voltage. This phenomenon allows for the efficient implantation of metal ions onto the surface of alumina ceramics. We have shown that the sheet resistance of dielectric targets depends on the material of the implanted metal ions and decreases with an increase in the implantation dose by 3–4 orders of magnitude compared with the initial value at the level of $10^{12}$ $\Omega$ per square.

**Keywords:** ion beam; metal ions; charge compensation; ion implantation; ceramic target; sheet resistance

## 1. Introduction

The interaction of beams of accelerated ions with targets made of electrically insulating materials may lead to the charging of their surface up to a potential comparable to the accelerating voltage [1]. This is especially true in the case of the use of focused ion beams of small diameter, for example, in systems of secondary ion mass spectrometry [2] and technologies for precision ion beam exposure [3]. In this case, a significant part of the total ion flux directed to the target can be deflected by the electric field of the charged surface. Such losses can be minimized using methods for compensating the surface charge of the target, for example, by flooding with electrons the region of interaction of ions with the target surface [4], the preliminary deposition of a thin conductive coating on this surface, generating plasma near the treated surface [5], and a pressure increase in the region of the ion beam drift space [6]. The last two methods lead to the effect of gas ions on the treated surface from the residual atmosphere of the vacuum chamber, but for certain technological processes, it may be unacceptable. The generation of metal ion beams with an ultra-low proportion of gas ions becomes possible when using a source based on a vacuum arc discharge because they do not have a lower limit on working pressure [7,8]. However, at a residual pressure in the drift region of the metal ion beam, of the order of $10^{-6}$ Torr or less, the effect of beam plasma electrons on the compensation of the surface charge of the irradiated target and the space charge of the beam is negligible. It should be noted that during the treatment of targets with a conductive base with a thin dielectric film on the surface by accelerated ion beams, electrical breakdowns may occur in the volume of the

material of this film [9,10]. Such processes lead to the local destruction of such film coatings and a decrease in their quality.

In the case of solving the problem of surface charging, the implantation of metal ions in the surface of dielectrics is an effective tool for the synthesis of near-surface layers consisting of a composite of the target material and the material of the implanted impurity [11–13]. The result of the implantation process is a modification of the mechanical, optical and electrical properties of the dielectric surface while maintaining the original properties in the body of the dielectric [14,15]. Known studies are related to the implantation of metal ions in the surface of ceramic materials based on oxides of aluminum, titanium, magnesium, silicon, and nitride and boride ceramics [16]. In these works, one of the indicators of the successful implantation of metal ions in the surface of insulating materials is a decrease in surface resistance [17–20]. It is shown that the after effect of the implantation of metal atoms is the formation of a conductivity matrix in the near-surface layers, which consists of separate or overlapping islands formed by the introduced atoms. Conduction electrons are transported to the surface layer when an external electric field is applied as a result of hopping conductivity [21], percolation [22], direct passage along conduction channels, and also as a result of combination. There are known also works on the implantation of metal ions into polymeric materials [23] to increase their surface conductivity, increase wear resistance [24], and the formation of bactericidal properties on the surface [25]. Thus, the issue of the ion implantation of materials with dielectric properties and the solution to the problem of charging their surface with a beam has not only a scientific but also a practical aspect.

In a number of studies, the explanation is used that the compensation of the surface charge of the ion beam target can have a direct influence on the compensation of the space charge of the ion beam [26,27]. However, there are objective reasons showing that this is not enough. Compensated beams, first of all, facilitate the transport of ions to the target without significant losses. If the target is made of a dielectric or is metallic but is reliably insulated from the grounded parts surrounding it, and also if its area significantly exceeds the cross-sectional area of the ion beam, then it will be charged by this beam. A source of electrons is needed to neutralize this positive surface charge. Nevertheless, even in the absence of such a source, processes of effective interaction of ion beams with non-conducting materials take place, resulting in the presence of implanted atoms in the near-surface layer [28] whose ions were in the beam.

Thus, along with studies of the interaction of accelerated metal beams on insulated targets and the results of ion modification of their surface properties, the issues of determining the source of electrons that neutralize the charge introduced by the ion beam are topical, and these issues are the subject of this work.

## 2. Materials and Methods

The studies were carried out on an experimental stand equipped with a source of metal ions based on a vacuum arc discharge Mevva-5.Ru [29]. The outlines of the experimental setup in two configurations are shown in Figures 1 and 2. A vacuum arc discharge was operated between cathode 1 and hollow anode 3 with a current of up to 200 A, with a pulse duration of 250 μs, and with a repetition rate of 10 s$^{-1}$. The extraction of ions from the plasma of a vacuum arc was carried out through the holes of a multi-aperture ion–optical system formed using emission electrode 5, to which a high voltage of up to 50 kV of positive polarity was applied, and suppressor electrode 6, to which a negative voltage of up to 1 kV was applied to reflect secondary electrons knocked out from the extraction electrode 7, a result of secondary ion–electron emission. Extraction electrode 7 was under the ground potential. The extracted ion beam was transported in the equipotential space of a grounded vacuum chamber 8, which was made of stainless steel.

The diagnostics of the ion beam parameters were carried out using an experimental setup in the configuration shown in Figure 1. To measure the mass–charge composition of the ion beams, a time-of-flight mass–charge spectrometer was used, which was located

directly behind the collector–holder of the samples. As a result of the analysis of the obtained mass–charge spectra, the values of the average charge states of metal ions were determined in the case of using each of the cathodes.

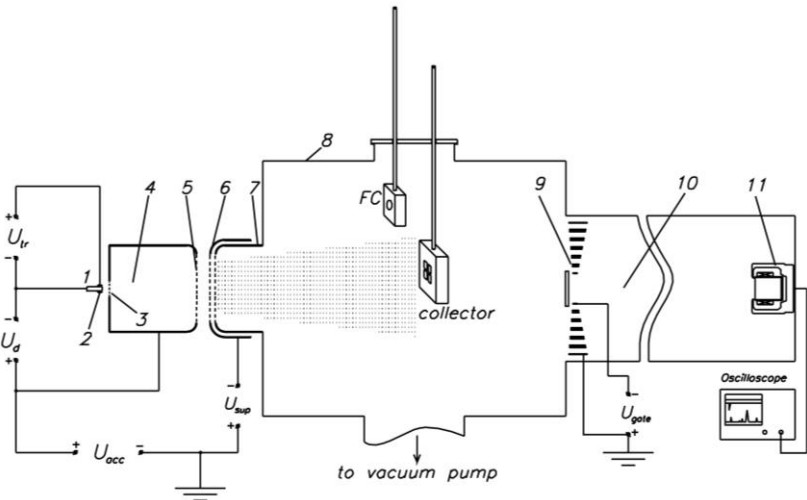

**Figure 1.** Scheme of the experimental setup: 1, vacuum arc discharge cathode; 2, initiating discharge anode; 3, anode diaphragm; 4, hollow anode; 5, emission grid; 6, suppressor grid; 7, extracting electrode (grounded); 8, vacuum chamber (grounded); 9, deflecting system of the time-of-flight spectrometer; 10, drift tube (grounded); 11, beam current sensor.

Concerning the experiments used to study the processes of charging non-conducting targets, the configuration of the experimental setup shown in Figure 2 was used. In this case, the time-of-flight spectrometer was removed from the vacuum chamber, and a collector displacement system was mounted in its place. Round metal collector 6, made of an aluminum sheet 1 mm thick and that was electrically insulated from the walls of chamber 7, was located on movable rod 10 coaxially with the vacuum chamber. This approach made it possible to directly measure the floating potential using a high-voltage voltage divider, depending on the distance between the collector and the extraction electrode of the ion–optical system. The inner diameter of the cylindrical vacuum chamber was 70 cm. The diameter of the ion beam collector was 55 cm. Thus, two conditions were met: ensuring the electrical strength of the vacuum gap between the collector and the walls of the vacuum chamber, as well as complete overlapping of the ion beam cross-section at moving the collector along the grounded vacuum chamber. The collector was attached to holder 8, which provided isolation from the grounded movable rod and the vacuum chamber and also served as a mechanical fastening for the electrical contact with the upper arm of the high-voltage voltage divider. To measure the floating potential of an insulated collector, a Tektronix P6015A measuring probe 12 with a division ratio of 1:1000 was used.

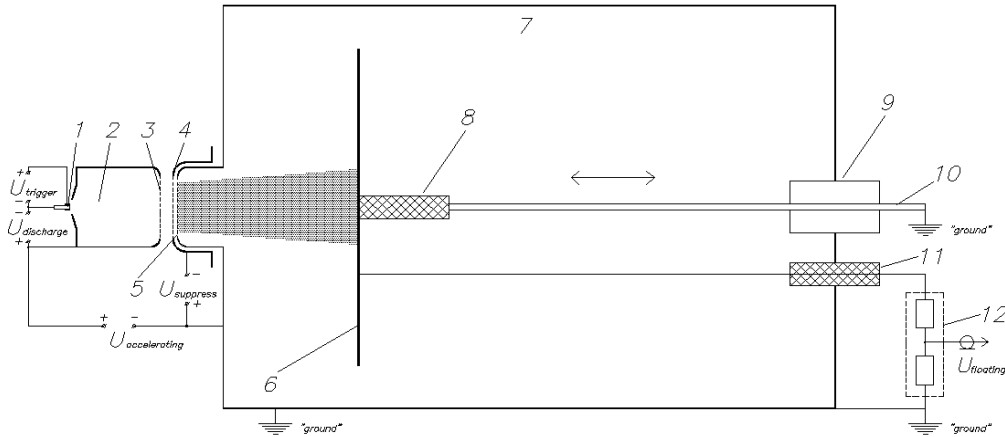

**Figure 2.** Scheme for measuring the floating potential of an insulated collector. 1, vacuum arc cathode; 2, hollow anode; 3, emission electrode; 4, suppressor electrode; 5, extracting electrode (grounded); 6, insulated collector; 7, vacuum chamber; 8, high-voltage collector holder insulator; 9, vacuum seal; 10, collector holder (grounded); 11, high-voltage input of the voltage divider; 12, high-voltage voltage divider.

To study the effect of the accelerated metal ion beams interaction with the dielectric targets, the method of measuring the surface resistance was used. Ions of the electrically conductive materials are implanted onto the surface of alumina ceramic samples: carbon, C; titanium, Ti; gold, Au; platinum, Pt; and tantalum, Ta. The distance between the emission electrode of the ion source and the beam collector was about 60 cm. The experimental samples were made of alumina ceramics. Plates 1 × 1 cm in size and 1 mm thick were glued with the reverse side of a double-sided adhesive tape with an aluminum base to a water-cooled aluminum collector–holder for more efficient heat dissipation. Thus, the temperature of the samples during implantation did not exceed 40 degrees Celsius. High-temperature annealing of the samples was not performed.

To measure the implantation distribution profiles, a PHI 6300 secondary ion mass spectrometry setup (Perkin-Elmer PHI 6300 Ion Microprobe, Waltham, Massachusetts, USA) was used. The sputtering of surface atoms was carried out using a source of Cs+ ions with an energy of 7 keV. The surface of the samples was scanned using a focused ion beam. The scanning area was a rectangle with sides of 500 μm. In order to eliminate the edge effect of the crater for analysis using a diaphragm, secondary ions coming from the central part of the etch crater (9% of the total crater area) were collected. To neutralize the positive charge accumulated on the sample surface during analysis, an electron gun was used. After measurements, the depth of the etch crater was determined using a profilometer, and then the dependences of the intensity of secondary impurity ions on the etch depth were plotted.

The surface resistance of alumina ceramic samples was measured using an E6-13A teraohmmeter manufactured by Punane-Ret, Таллин, Estonia.

## 3. Results and Discussion

The parameters of the metal ion beams generated by a source based on a vacuum arc discharge were studied using the experimental setup configuration shown in Figure 1. In this case, a Faraday cup was used as a collector with the possibility of its movement along the ion beam cross-section at a distance of 60 cm from the extracting electrode of the ion–optical system. Its main element is a cylindrical copper cup, which was electrically connected to the "ground" with a shunt resistance of 1 kOhm, and which collected the ion flow cut out from the beam through a diaphragm with a hole of 1 cm². A magnetic field with an induction of approximately 0.01 T transverse to the direction of ion motion, created through the use of permanent magnets, prevented the escape of electrons knocked

out of the collector as a result of ion-induced secondary electron emission. As a result, the trajectory of the electrons was such that they returned to the walls of the copper cup. Thus, the distribution profile of the current density over the beam cross-section was studied without the influence of secondary ion–electron emission (Figure 3).

A time-of-flight mass–charge spectrometer [30] was mounted downstream of the ion beam behind the moving Faraday cup at the end of the vacuum chamber. The operating principle of the spectrometer is as follows. When a voltage pulse with a duration of 80 ns and an amplitude of 4 kV was applied to the gate plates (Figure 1, pos. 9), the beam ions were deflected by a small angle to the center of the drift tube (Figure 1, pos. 10). Since the duration of the deflecting pulse is much shorter than the time of flight of ions $\tau$ of the base of the spectrometer $L$, in the course of the movement of ions from the gate to the Faraday cup of the spectrometer, the ion beam components with different values of $M_i/Q$ were separated into groups during the movement. In this case, peaks were observed in the circuit for measuring the current of the Faraday cup, corresponding to the time $\tau$ of reaching the working surface of the Faraday cup by ions with a certain value of $M_i/Q$.

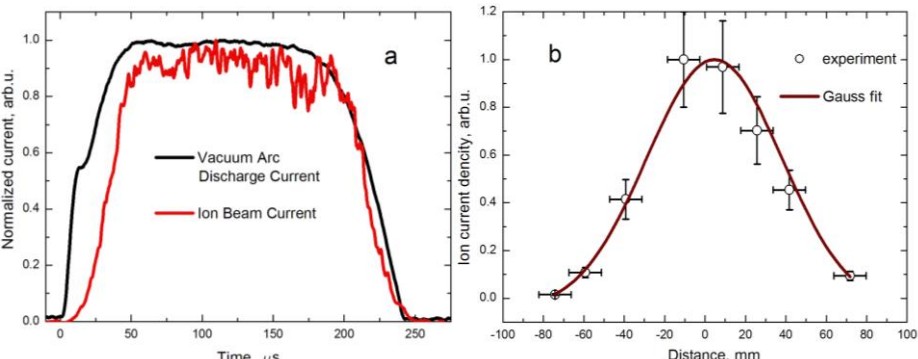

**Figure 3.** Vacuum arc discharge current with a copper cathode and ion beam characteristics: (**a**) waveforms of the vacuum arc discharge current and copper ion current at the center of the beam, normalized to maximum values, $I_d = 200$ A, $I_i = 4$ mA; (**b**) the current density distribution over the ion beam cross-section normalized to the maximum value of 4 mA/cm$^2$.

The time-of-flight $\tau$, defined as the delay between the application of a deflecting pulse to the spectrometer shutter and the signal in the Faraday cup circuit, is related to the charge and mass of the ions, obtained using the following expression:

$$\tau = L \cdot (M_i/Q)^{1/2} \times [1/(2 \cdot e \cdot U)]^{1/2} \tag{1}$$

From the value of $\tau$ recorded using the oscilloscope, the ratio $M_i/Q$ of ions was determined, and the proportion of ions of each charge was calculated by integrating the corresponding current peaks. In this case, the peak amplitude of the current pulse for ions with charge $Q$ was $Q$ times higher than for the singly charged.

The results of studying the mass–charging composition of the ion beams extracted from the plasma of a vacuum arc discharge with cathodes made of graphite, titanium, copper, gold, platinum, and tantalum are shown in Figure 4 as time-of-flight spectra normalized to the amplitude of the maximum peak. Under the symbols of ions with the charge state $n+$, the values of their fractions $f_n$ in the total particle flux of the corresponding material are given, as well as the values of the average charges <$Q$>, determined using the following expression (2).

$$<Q> = \Sigma n \cdot f_n \tag{2}$$

where $f_n$ is the ion fraction which corresponds a charge state $n = 1 \div 6$.

The relatively high purity of ion beams should be noted. Impurities in the form of singly charged ions of atomic oxygen, molecular and atomic hydrogen, and carbon make up a small percentage of the total flux of the charged particles in the beam. An exception

was the case when a graphite cathode was used, and carbon ions were the main particles in the beam, and their fraction was 100%. The presence of metal ions with charge states of up to $Q = 3+$, obtained with the use of titanium and gold (Figure 4b,d, respectively), up to $Q = 4+$, in the case of copper and platinum (Figure 4c,e, respectively), and up to $Q = 6+$ for tantalum (Figure 4f), are typical situations [8], determined by the nature of the functioning of the vacuum arc discharge and the properties of the cathode materials [31–33]. The average charge state of ions in the vacuum arc plasma $<Q>$ for a certain cathode material remains constant in a wide range of discharge currents, from several units of amperes to several hundred amperes, and does not depend on the accelerating voltage. In turn, the average energy of the $E_{mean}$ ions that have passed through the accelerating gap of the ion––optical system is determined by multiplying the average charge $<Q>$ and the accelerating voltage $U_{acc}$ (3).

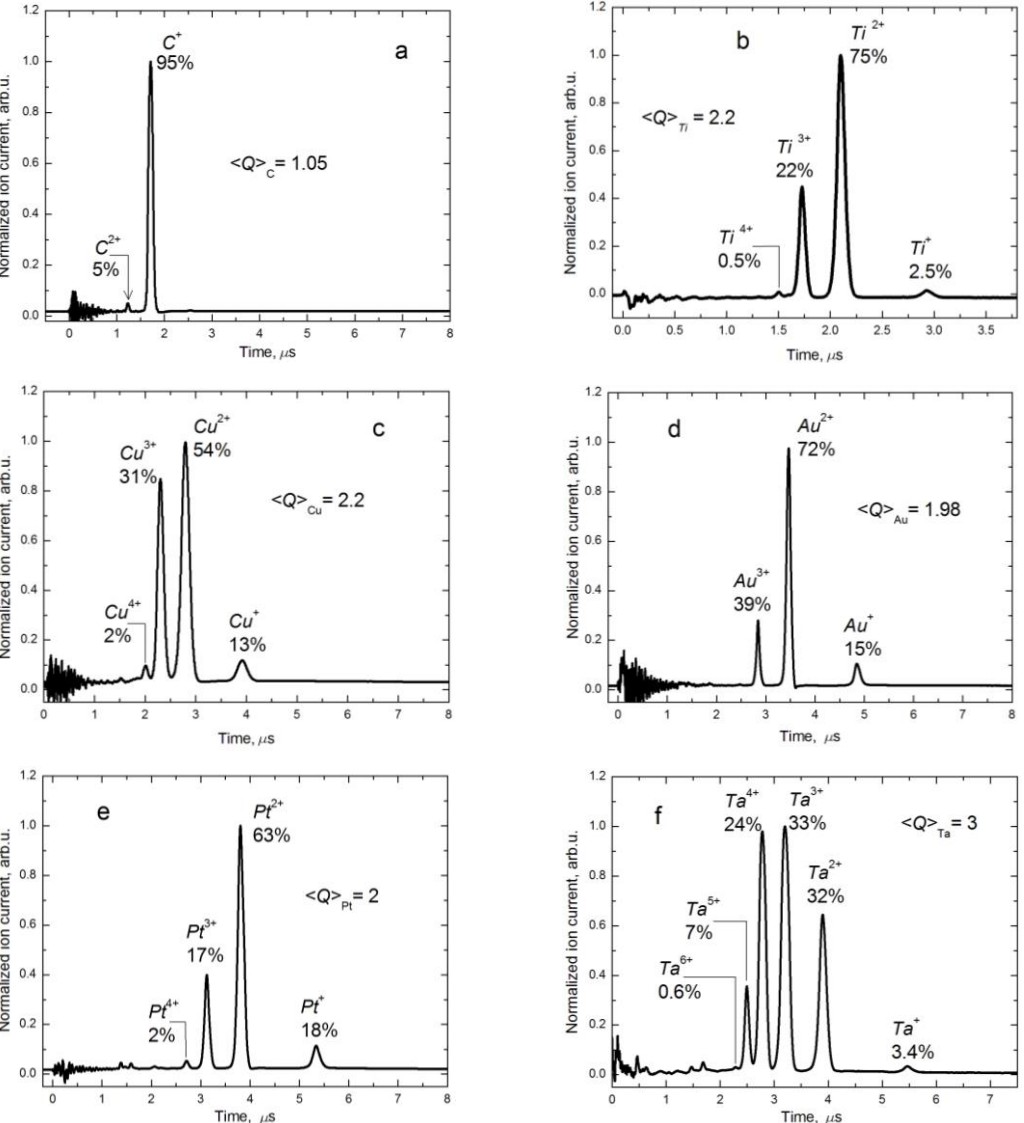

**Figure 4.** Time-of-flight spectra of ion beams recorded in the case of using vacuum arc discharge cathodes: (**a**) graphite @ 21 kV (here and after given value of accelerating voltage); (**b**) titanium @ 29 kV; (**c**) copper @ 22 kV; (**d**)—gold @ 44.3 kV; (**e**) platinum @ 35.5 kV; (**f**) tantalum @ 31.2 kV.

$$E_{mean} = <Q> \cdot U_{acc} \tag{3}$$

When studying the dependence of the floating potential of an insulated collector, a copper cathode with a vacuum arc discharge was used, and, consequently, a beam of copper ions with an average charge of $<Q>_{Cu}$ = 2.2. The complete coverage of the ion beam by the collector was confirmed experimentally. In the entire range of the longitudinal movement of the collector grounded through a shunt resistance, from 5 to 60 cm from the extraction electrode of the ion–optical system, the total beam current did not depend on the distance. Figure 5 shows the waveforms of the potential of an insulated collector obtained at a minimum pressure of $1 \times 10^{-6}$ Torr and an accelerating voltage of 30 kV. The numbers of the waveforms correspond to a certain distance from the ion–optical system of the ion source.

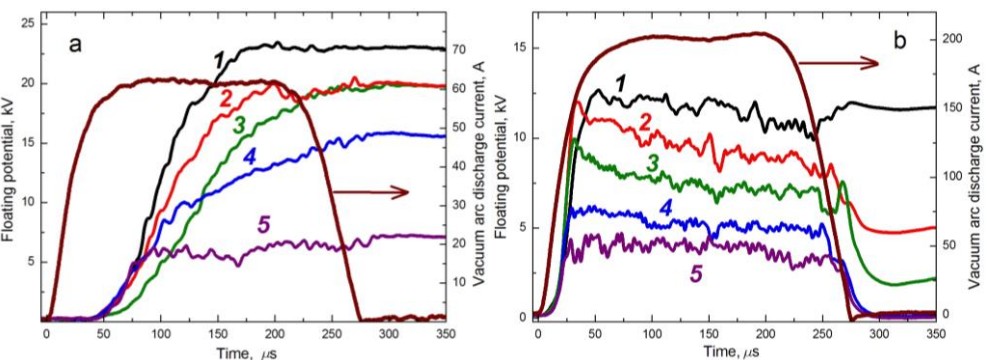

**Figure 5.** Waveforms of the floating potential of the insulated collector and the current of the vacuum arc discharge: (**a**) $I_d$ = 60 A; (**b**) $I_d$ = 200 A; 1—5 cm, 2—10 cm, 3—20 cm, 4—30 cm, 5—40 cm—collector position at both $I_d$.

At a vacuum arc discharge current amplitude of 60 A and a distance of 5 cm, the potential value was about 23 kV (Figure 5a). Removing the collector to a distance of 40 cm from the extracting electrode led to a decrease in the potential by more than three times. With an increase in the vacuum arc current to 200 A, an insulated collector located in the same places of the inner space of the vacuum chamber in the path of the ion beam (Figure 5b) was charged to potential values almost two times lower than $I_d$ = 60 A. At such a low pressure, ionization processes in the ion beam drift region can be neglected, and the possibility for the charge compensation of the ions by beam plasma electrons is negligible. The most possible reason for the decrease in potential is the emission of electrons from the walls of the vacuum chamber and the electrodes of the ion–optical system under the influence of the ions reflected by the electric field of a charged collector. Removing the isolated target from the ion source increases the area of the inner surface of the vacuum chamber with which the reflected ions interact. Thus, the number of electrons emitted as a result of secondary ion–electron emission increases. For these electrons, the field of the positively charged collector is accelerating. Accordingly, the electron flow partially compensates for the positive charge introduced by the metal ion beam, resulting in lower values of floating collector potential. An increase in the current of the vacuum arc discharge at the same value of the accelerating voltage, led to a mismatch between the plasma concentration in the hollow anode and the electric field in the multi-aperture electrode system for extracting ions [34]. This caused significant losses in terms of the ion beam on the electrodes and a sharp increase in the ion-induced emission of electrons, both from the walls of the vacuum chamber and from the surface of the electrodes of the ion–optical system, and, accordingly, to a more effective decrease in the potential of the isolated collector.

Increasing the pressure in the area of the extraction and transport of the ion beam through argon feeding also led to a decrease in the floating potential. However, as follows from the analysis of the waveforms, as presented in Figure 6, the decrease in potential, in this case, has a different character. It is obvious that simultaneously with the process of compensation of the collector charge by secondary electrons, interaction with beam

plasma electrons also occurs, the formation of which at a pressure of the order of $10^{-4}$ torr is quite effective. This makes it possible to significantly reduce the potential of an isolated collector to values at the level of several hundred volts (Figure 6b).

To evaluate the efficiency of the interaction of an ion beam with a nonconductive target, the SIMS method was used to study the elemental composition of the surface of alumina ceramic samples after the implantation of titanium ions at an accelerating voltage of 30 kV with an exposure dose of $1 \times 10^{16}$ ion/cm$^2$.

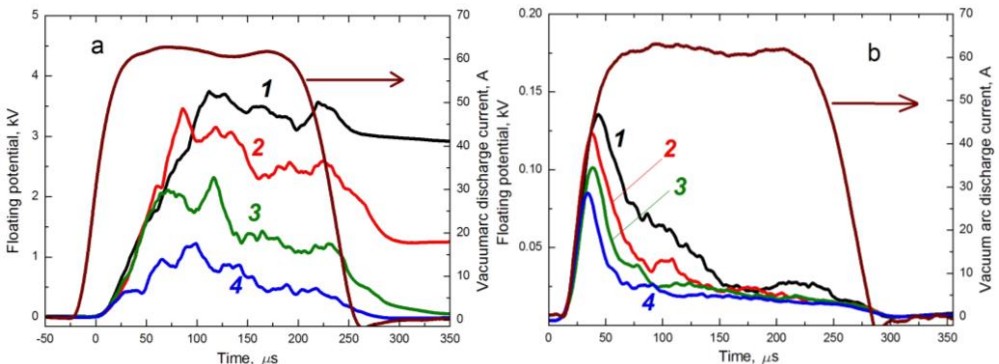

**Figure 6.** Waveforms of the floating potential of an insulated collector and the current of a vacuum arc discharge with an additional puff of argon: (**a**) $p_1 = 6 \times 10^{-5}$ torr; (**b**) $p_2 = 4 \times 10^{-4}$ torr; 1–5 cm, 2–10 cm, 3–20 cm, 4–40 cm—collector position at both pressures.

As a result of the analysis of the mass spectra of the sputtered and ionized atoms, it was shown that titanium atoms were present among the sputtering products. Their depth distribution is shown in Figure 7. On the surface of the alumina ceramics, the proportion of titanium atoms is slightly more than $5 \times 10^{-3}$ at.%. In the direction from the surface of the substrate towards its body, there is an almost linear increase in the content of titanium atoms up to a level of $4 \times 10^{-2}$ at.% at a depth of approximately 13 nm. After this mark, the distribution function increases non-linearly until the maximum value of $7 \times 10^{-2}$ at.% is reached at a depth of 48 nm. As follows from Figure 7, the "hump" of the distribution is weakly expressed and rather resembles a "plateau", which occupies the region from 23 to 60 nm deep from the sample surface. The etch depth of the diagnostic crater was approximately 160 nm. Up to this value, the distribution function shows a gradual decrease in the content of titanium atoms to a level of $2.7 \times 10^{-2}$ at.%.

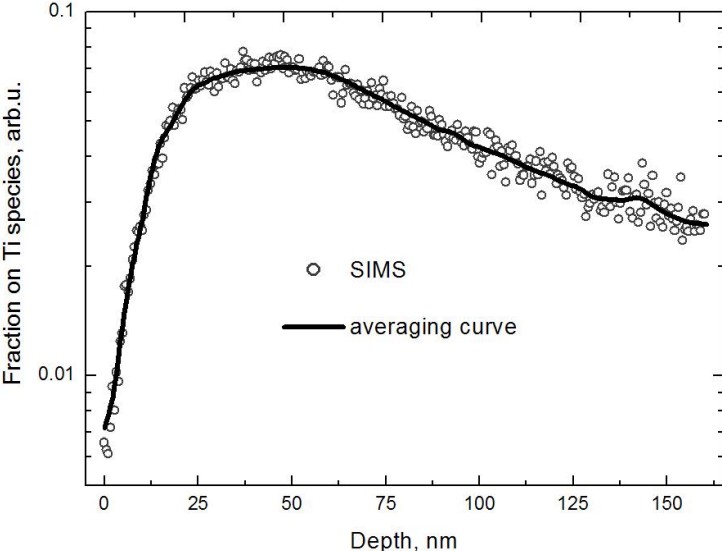

**Figure 7.** Depth distribution profile of titanium atoms implanted in alumina ceramic measured by SIMS ($E_{mean} = 66$ keV, $D_i = 1 \times 10^{16}$ ion/cm$^2$).

The surface resistance of the samples of alumina ceramics after the implantation with an ion beam was studied. A special fixture was used, which was formed using the insulating plates from the original alumina ceramics, a ceramic sample subjected to ion modification with foamed graphite electrodes superimposed on its surface. This sandwich-like structure was fixed on both sides with clerical clips, as shown in Figure 8.

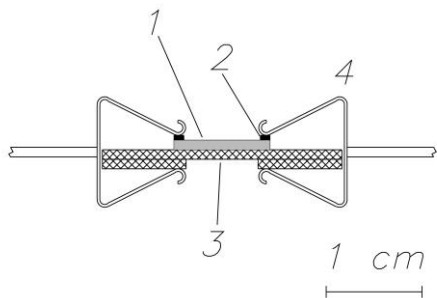

**Figure 8.** System of electrodes for measuring surface resistance: 1, experimental sample; 2, contact plates (made from foamed carbon gasket); 3, ceramic insulator; 4, fixing clips.

The sheet resistance $R_s$ was obtained on the basis of the resistance R, which was measured using the teraohmmeter. Using the relation (4),

$$R = R_s \cdot (l/w) \tag{4}$$

where R is the measured resistance, $R_s$ is the sheet resistance, $l$ is the distance between the electrodes, and $w$ is the electrode's width. The sheet resistance $R_s$ is determined by multiplying the measured resistance R and the ratio of the width of their contact with the surface $w$ to the length of the gap $l$ between these contact electrodes (5).

$$R_s = R \cdot (w/l). \tag{5}$$

Typical values of these parameters were $l = 5 \pm 1$ mm and $w = 8 \pm 1$ mm, respectively.

The choice of the dimension of the surface resistance deserves special attention. Traditionally, "Ohm·m" or "Ohm·cm" are used, and to switch to these units of measurement, it is necessary to multiply the surface resistance determined using expression (5) by the thickness of the ion-modified near-surface layer. This approach is used to determine the surface resistance of deposited coatings, provided that the thickness and the elemental and phase composition of the coating are highly uniform. However, because of the analysis of the implantation profile obtained using secondary ion mass spectrometry, it was shown that the distribution of the concentration of implanted metal atoms over depth is nonuniform (see Figure 7). Therefore, under the conditions of this study, the use of the unit of measurement of surface resistance, "Ohm·cm", would not be correct. Connected to this fact, the dimension of "Ohms per Square" was used, the value of which is directly determined from (5).

The surface resistance of the ceramic substrates after the implantation as a function of the ion implantation dose $D_i$ is shown in Figure 9. It can be seen that the surface resistance decreases with increasing implantation dose. This unambiguously indicates the effectiveness of the use of metal ion implantation to create conductivity in the ceramic surface layer. We assume that this phenomenon is caused by the passage of electrons through the composite of the target materials and the implanted metal in the surface layer, in other words, small conductive regions immersed in an insulating medium. The formation of electrical conductivity in such composites is described by the theory of percolation [35], one of the statements of which is the criterion for an increase in conductivity when the threshold concentration of the conducting elements in the "conductor-insulator" system is reached. The data presented in Figure 9 also show that for the same implantation dose, the surface resistance depends on the material of the implanted ions.

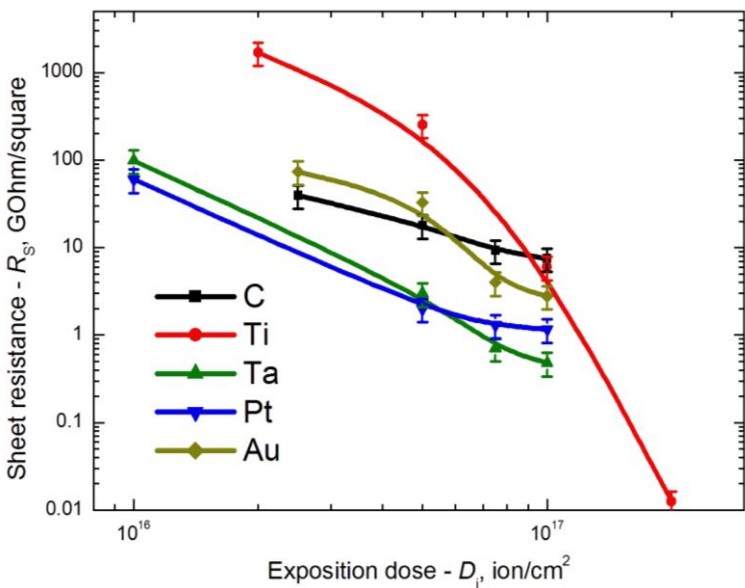

**Figure 9.** Surface resistance vs. ion implantation dose for a number of different ion species (alumina ceramic substrate, ion energy: 52.5–150 keV).

## 4. Discussion

Based on the results presented in Figure 5a, it can be concluded that for the wide-aperture ion beam generated using the source based on the vacuum arc discharge, in this study, the conditions were always formed to compensate for the space charge of the ion beam. This means that a significant fraction of the ions in the beam reached the collector under high vacuum conditions, even if the collector was at a floating potential. When such a collector was located in the immediate vicinity of the extracting electrode of the ion source, the ions charged it to a potential level of 75% of the accelerating voltage (22.5 kV, at $U_{acc}$ = 30 kV). This was a consequence of the features of the source of the metal ions Mevva-5.Ru operating. The point is that it is extremely difficult to achieve full agreement between the plasma concentration in the hollow anode of the discharge system and the electric field in the accelerating gap of the ion–optical system. Basically, the plasma partially penetrates the accelerating gap, and the extraction of ions is carried out from the open plasma boundary. This leads to defocusing of ion fluxes passing through most of the holes in a multi-aperture three-electrode ion–optical system. Accelerated ions, directed at an angle to the central axis of the source, hit the electrodes of the ion–optical system and the wall of the cylindrical vacuum chamber, from which electrons were knocked out as a result of ion-induced electron emission [36]. The electric field between the grounded chamber and the space charge of the ions accelerated the secondary electrons in the direction of the ion beam space. If the imbalance between the plasma concentration and the accelerating voltage in the ion source increased, for example, with an increase in the vacuum arc discharge current by several times (Figure 5b), then the floating potential of an insulated collector located at the same distance from the ion source decreased dramatically and was no more than 30% of the accelerating voltage.

The reason for the decrease in the floating potential of an insulated collector (Figure 10) when it is removed from the ion source was as follows. When the insulated collector was reached by the ion flux at the front of the ion current pulse, it was charged to a sufficiently high positive potential. The subsequent spatiotemporal group of ions in the pulse experienced the decelerating effect of the field of a similarly charged collector. The slowed ions were reflected by the electric field of the beam toward the walls of the vacuum chamber, where they arrived with an energy sufficient to initiate ion–electron emission. These secondary electrons, according to the scenario described above, were accelerated toward the ion beam and the charged collector. Thus, partial compensation of the charge brought

about by the accelerated ions took place. When the collector moved away from the ion source, there was an increase in the surface area of the cylindrical wall of the vacuum chamber (Figure 2), which surrounded the space of the ion beam between the ion source and the collector. The consequence of this was an increase in the flux of secondary electrons, contributing to an even greater neutralization of the space charge of the beam and the charge of the collector. An increase in the pressure in the region of beam transport led to a decrease in the collector potential to a level of 10% of that of the accelerating voltage. In this case, the accelerated metal ions collided with argon atoms and experienced the processes of recharging and deceleration. In this way, pairs were generated—fast neutral atoms and slow ions—which were accelerated by the space charge field of the ion beam toward the wall of the vacuum chamber, from which electrons were knocked out. These electrons filled the ion beam drift space and compensated for the collector charge. The fact that the ion recharging took place in a fairly extended volume and not only in the region of a charged collector led to the generation of a larger flux of reflected ions and, consequently, a larger flux of compensating electrons. Therefore, for example, when the pressure was increased to $4 \cdot \times 10^{-4}$ Torr, the insulated collector was charged at the front of the beam current pulse to a potential of no more than 150 V at an accelerating voltage of 30 kV, which dropped during the pulse duration to a level of 10 V (Figure 6b).

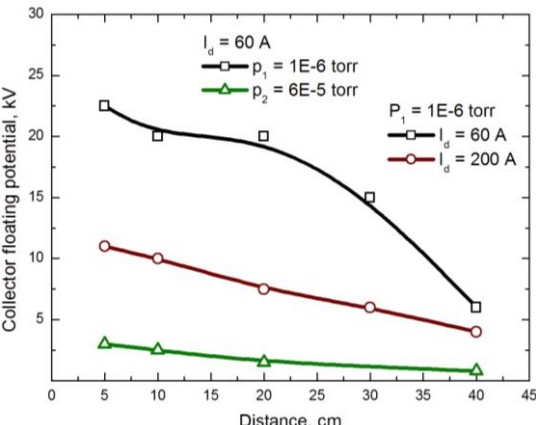

**Figure 10.** Dependences of the floating potential of an isolated collector (pulse plateau) on the distance to the ion source (data are taken from Figures 4 and 5 in this paper).

The above reasoning is confirmed through the experimental facts of the successful ionic modification of the surface properties of the insulating materials [17–20], including in relation to the present study. The study of the implantation profile using the SIMS method (Figure 7) is an indirect confirmation of the conditions required for the removal of the charge from the surface of the alumina ceramic during implantation.

A significant part of the research in this area is focused on the creation of low-conductivity near-surface layers of ceramic insulators. In this direction, a number of results have been obtained that require analysis and generalization. The obtained dependencies of the surface resistance on the implanted dose using some ion materials (Figure 9) demonstrate a general trend of decreasing surface resistance with increasing implantation dose. However, the influence of the material of the implanted ions during the formation of the conductive layer does not yield an explicit monotonic dependence, but there is an assumption that it is possible to streamline the data, arranging them in accordance with the order of increasing atomic mass. In accordance with this assumption, the dependencies of the surface resistances obtained at implantation doses of $2.5 \times 10^{16}$, $5 \times 10^{16}$, $7.5 \times 10^{16}$, and $1 \times 10^{17}$ ion/cm$^2$ on the atomic mass of a number of implanted ion materials: C, Mg, Ti, Zn, Zr, Sn, Ta, Pt, Au, and Pb [37] (Figure 11).

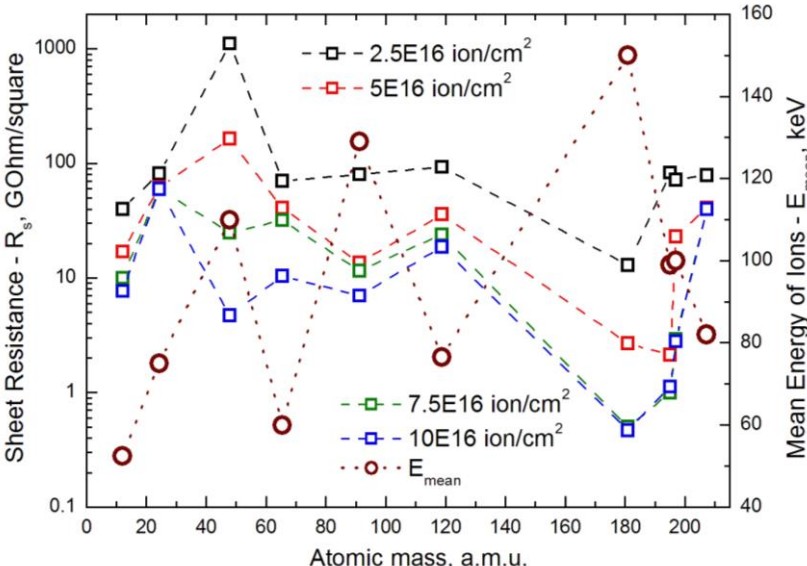

**Figure 11.** Sheet resistance for a number of implantation doses and mean energy of ions in the ion beams vs. atomic mass.

The dotted lines between the points are drawn conditionally, since it was not possible to study the full range of materials. Nevertheless, at implantation doses of more than $5 \times 10^{16}$ ion/cm², the positions of the experimental points acquire an identical structure. It is quite obvious that the surface resistance tends to increase on going from tantalum to platinum, gold, and lead. On this coordinate field (Figure 11), the dependence of the average energy of accelerated ions was also plotted, which was determined based on time-of-flight mass–charge spectrometry data (Figure 4) and [8] using Equation (3), where $U_{acc}$ = 50 kV in all experiments on ion implantation. When compared with the dependencies of the surface resistance obtained at implantation doses of more than $5 \times 10^{16}$ ion/cm², it is noticeable that, starting from titanium (atomic mass—47.9 a.m.u.), the experimental points are in antiphase. The local maximum of the average ion energy corresponds to the local minimum of the surface resistance, and vice versa. Of course, the results of ion implantation do not store the memory of the charge state of metal ions in the beam. Nevertheless, the kinetic energy of ions interacting with the target material ultimately determines the configuration of the implantation profile in the surface layer [38,39], which determines the conditions for transporting charges in a composite medium formed by metal particles immersed in the insulating material of the ionic target. For a more complete physical understanding of the surface resistance, further studies will be required to fill in the gaps in the dependencies presented in Figure 11.

## 5. Conclusions

To neutralize the positive charge of an isolated target, the nature of the appearance of electrons is not important. However, at pressures less than $5 \times 10^{-5}$ Torr, the role of the electrons emitted from the walls of the vacuum chamber and the electrodes of the ion beam extraction system as a result of ion-induced electron emission is decisive. Therefore, when the collector is removed to a considerable distance, the surface area with which the beam ions interact, deflected by the electric field of the charged insulated collector, is many times greater than that of the surface area of the emission electrode, and the effect of compensating the collector charge by the flow of secondary electrons appeared to a greater extent. The obtained results explain the practical possibility of implanting metal ions onto the surface of targets based on dielectric materials, even in the absence of special external electron sources, for the removal of the charge from the treated surface.

The surface resistance of alumina ceramics, formed as a result of the implantation of metal ions, is determined through a combination of parameters, such as the material of the ions, the implantation dose, and the energy of the implanted ions. By selecting these parameters, it is possible to create materials with unique electric properties in terms of the surface.

**Author Contributions:** Conceptualization, K.P.S. and E.M.O.; methodology, K.P.S., E.M.O., and G.Y.Y.; validation, K.P.S., E.M.O., and G.Y.Y.; formal analysis, K.P.S., E.M.O., and G.Y.Y.; investigation, K.P.S., A.G.N., and G.Y.Y.; resources, K.P.S., E.M.O., and G.Y.Y.; data curation, K.P.S.; writing—original draft preparation, K.P.S., E.M.O., and G.Y.Y.; writing—review and editing, K.P.S., E.M.O., G.Y.Y., and A.G.N.; visualization, K.P.S. and E.M.O.; supervision, A.G.N. and E.M.O. All authors have read and agreed to the published version of the manuscript.

**Funding:** This research was funded by the state task of IHCE SB RAS project FWRM-2021-0006.

**Institutional Review Board Statement:** Not applicable.

**Informed Consent Statement:** Not applicable.

**Data Availability Statement:** Not applicable.

**Acknowledgments:** The authors greatly appreciate Ian Brown (Lawrence Berkeley National Laboratory) for helpful discussion as well as for English correction. Additionally, the authors consider it their duty to express gratitude to Alexander Chernyavskii (National Research Tomsk Polytechnic University, Tomsk, Russia) for providing SIMS diagnostics.

**Conflicts of Interest:** The authors declare no conflict of interest.

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
