# Peer review of "Neutralization of the Surface Charge of an Insulated Target under the Interaction of High-Energy Metal Ion Beams"

_qubs, doi:10.3390/qubs7020017_

Round 1
Reviewer 1 Report
The article deals with high voltage ion sources based on arc discharge and how they affect the charge distribution on ceramic dielectric materials. The article reports the experimental methods with details and presents extensive measurements various ions at different charge states. I found the "discussion" section somehow overly long, but it is probably fine as is. I recommend publication of the article as is.
Minor comments
- line 318 "Ohm/ <unreadable symbol>"
- line 367. The authors refer to a "coefficient", but is is not fully clear to me what the coefficient stands for. Maybe they refer to the number of secondary electron emitter per impinging ion.
Author Response
Thanks for the comments.
Point 1.
An unreadable symbol has been replaced with the word "Square".
Point 2.
The sentence has been reduced to the following form: "Slowed ions were reflected by the electric field of the beam towards the walls of the vacuum chamber, where they arrived with energy sufficient to initiate ion-electron emission".
We guess it is clear enough expression of idea.
Please see for attached file. All corrections are pointed by yellow background.

Reviewer 2 Report
Ion-surface interactions in implantation certainly depend on the conductivity of the target material and may be modified by auxiliary electrodes, electron beams, or low-pressure background gas, and probably more. This field has been studied since decades, and as in many materials and plasma studies, the complexity of the actual situation has turned out to be almost overwhelming. Further experimental work trying out additional approaches and addressing new details or compositions is certainly needed in order to eventually gaining insight and establishing systematics. In this sense, the present investigation is a welcome addition to the field and to the QBS journal.
Clearly the authors are on top of the material they present. The research is undertaken competently and described sensibly.
In line 406, a Kyrillic character has entered a mathematical expression and needs to be explained or replaced.
The language of the manuscript overall is fine. However, "works" might better read as "studies". There are more details that might profit from a native English speaker with the proper topical knowledge clarifying some sentences language-wise, but that is not essential.
Author Response
Thanks for the valuable comments.
Please see attached file.
All fixes are highlighted in green.
